# Understanding Micronutrient Access through the Lens of the Social Ecological Model: Exploring the Influence of Socioeconomic Factors—A Qualitative Exploration

**DOI:** 10.3390/nu16111757

**Published:** 2024-06-04

**Authors:** James Stavitz

**Affiliations:** Athletic Training Education, College of Health Professions and Human Services, Kean University, Union, NJ 07083, USA; jstavitz@kean.edu

**Keywords:** micronutrient access, socioeconomic factors, dietary behaviors, qualitative research

## Abstract

Background: Micronutrient deficiencies disproportionately affect various populations, influenced by a complex interplay of socioeconomic factors. This study delves into the intricate relationship between socioeconomic status and micronutrient access, emphasizing the perceptions of affordability, availability, and the impact of social support networks. Methods: A qualitative research design was employed, using purposive sampling to enlist a diverse cohort of participants from varied socioeconomic backgrounds. The methodology comprised semi-structured interviews and focus groups to gather detailed insights into the participants’ experiences and views on micronutrient access. The analysis framework was based on the Social Ecological Model (SEM), enabling an in-depth examination of individual, interpersonal, community, and societal influences. Results: With 30 participants, aged 20–70, representing a range of unique characteristics such as differing health conditions, cultural backgrounds, and economic statuses, the study uncovered five key themes: Individual-Level Factors, Interpersonal Relationships, Community Environment, Societal Factors, and Intersectionality. These themes illustrate how personal dietary habits, social networks, community resources, and broader socioeconomic policies converge to shape micronutrient access, emphasizing the complex interplay of overlapping social identities and structural barriers. Conclusion: The findings highlight the need for holistic nutrition interventions that account for the extensive spectrum of socioeconomic determinants. This study not only enriches the theoretical underpinnings of the SEM but also provides actionable insights for crafting targeted strategies to enhance micronutrient access and ameliorate dietary inequities. It advocates for comprehensive public health initiatives attuned to the nuanced needs and hurdles faced by diverse demographic sectors.

## 1. Introduction

Micronutrients, encompassing essential vitamins and minerals, are fundamental for maintaining optimal health and preventing various diseases [1]. Access to and the consumption of foods rich in essential micronutrients are influenced by a myriad of socioeconomic factors (SEFs), which, in turn, contribute to significant disparities in health outcomes across different population groups [2]. Understanding the complex relationship between socioeconomic status (SES) and micronutrient access is paramount for addressing nutritional inequalities and promoting public health. Previous research has highlighted the association between SES and dietary patterns, indicating that lower SES is often linked with limited access to nutritious foods and a higher prevalence of food insecurity [3]. Individuals from lower SES backgrounds may perceive healthy foods, particularly those rich in essential micronutrients, as more expensive and less accessible compared to unhealthy alternatives [4]. Economic constraints often force individuals to prioritize cheaper, calorie-dense foods over nutrient-dense options, contributing to suboptimal dietary intake [5]. Additionally, social support networks play a crucial role in shaping dietary behaviors and nutritional practices [6]. Communities with strong social ties may facilitate the sharing of food resources, cooking skills, and nutritional knowledge, thereby enhancing individuals’ ability to maintain adequate dietary diversity and nutrient intake [7].

Despite existing research documenting the associations between SEFs and dietary habits, comprehensive qualitative investigations that delve into the lived experiences, perceptions, and behavioral patterns underlying micronutrient access within diverse socioeconomic contexts remain needed. Qualitative approaches offer valuable insights into the complex interplay between SES and dietary practices, providing a deeper understanding of the barriers and facilitators of micronutrient access. This study aims to address this gap by exploring how SEFs impact access to and the consumption of foods rich in essential micronutrients, with a specific focus on perceptions of affordability, availability, and the role of social support networks. By employing qualitative research methods, including interviews and focus groups, we aim to elucidate the lived experiences and challenges faced by individuals from different socioeconomic backgrounds in obtaining and incorporating essential micronutrients into their diets. Through rigorous analysis and interpretation of qualitative data, this study endeavors to inform the development of targeted interventions and policies aimed at promoting equitable access to vital nutrients and improving health outcomes across diverse populations. Furthermore, the role of social support networks, including family, friends, and community resources, in shaping dietary behaviors and nutritional practices cannot be understated. Social networks can serve as instrumental, informational, and emotional support sources, influencing individuals’ access to and utilization of resources for obtaining essential micronutrients [6,7]. For instance, social ties within communities may facilitate the sharing of food resources, cooking skills, and nutritional knowledge, thereby enhancing individuals’ ability to maintain adequate dietary diversity and nutrient intake [8,9].

### 1.1. Theoretical Framework: Social Ecological Model

The Social Ecological Model (SEM) provides a comprehensive framework for understanding the complex interplay of individual, interpersonal, community, and societal factors influencing access to and consumption of foods rich in essential micronutrients [10]. Developed by Bronfenbrenner (1979), the SEM posits that health behaviors are shaped by interactions across multiple levels of influence, ranging from personal characteristics to broader environmental contexts [10]. In the context of our study, the SEM serves as a guiding lens to systematically explore the determinants of micronutrient access and consumption among individuals from diverse socioeconomic backgrounds.

Individual Level: At the individual level, SES, education level, nutritional knowledge, attitudes, and beliefs about food and health are pivotal in shaping dietary behaviors [2]. Individuals’ perceptions of the affordability, availability, and nutritional value of foods influence their food choices and consumption patterns.Interpersonal Level: Social relationships and support networks, including family, friends, and community members, influence individuals’ dietary practices. Social support networks can facilitate or hinder access to nutritious foods, as well as shape individuals’ norms and behaviors related to food consumption [6].Community Level: The community environment encompasses neighborhood characteristics, access to food resources, and the availability of nutritional assistance programs. Factors such as the density of grocery stores, farmers’ markets, and food banks, as well as the presence of food deserts, influence individuals’ access to affordable, nutrient-rich foods [7].Societal Level: Structural and policy factors at the societal level, including economic policies, food assistance programs, agricultural subsidies, and marketing practices, shape the broader food environment. Socioeconomic policies and structural inequalities contribute to disparities in food access and nutritional outcomes among different socioeconomic groups [3].

Table 1 explains the SEM, depicting the interconnected levels of influence on individuals’ access to and consumption of foods rich in essential micronutrients. It highlights the reciprocal interactions between individual characteristics, interpersonal relationships, community environments, and societal factors, emphasizing the complex interplay shaping dietary behaviors and nutritional outcomes. Utilizing this theoretical framework, our study aims to provide a holistic understanding of the socioeconomic determinants of micronutrient access, thereby informing interventions and policies to promote equitable access to vital nutrients and improve health outcomes across diverse populations.

### 1.2. Overarching Research Question (RQ): The Overarching Research Question (RQ) for This Study on the Impact of SEFs on Micronutrient Access Is

How do SEFs influence access to and the consumption of foods rich in essential micronutrients, and what are the perceptions of affordability, availability, and the role of social support networks in shaping dietary behaviors?

The aim of the research question is to investigate the relationship between SEFs and access to essential micronutrients. Specifically, the question aims to

Explore how SES influences individuals’ ability to access and consume foods rich in essential micronutrients.Investigate perceptions of the affordability and availability of nutrient-rich foods among different socioeconomic groups.Examine the role of social support networks in facilitating or hindering individuals’ access to nutritious foods.Gain insights into the complex interplay between SEFs and dietary behaviors. The ultimate goal is to inform strategies to promote equitable access to vital nutrients and improve overall health outcomes.

## 2. Methodology

This general qualitative study utilized purposive sampling techniques to recruit participants, emphasizing diversity across cultural backgrounds, age groups, genders, and involvement in activities relevant to the topic under investigation. This approach facilitated the selection of participants capable of offering rich and varied insights relevant to the research objectives [11]. The recruitment process encompassed various steps, beginning with the identification of potential participants through existing networks, community organizations, and relevant online forums. Efforts were directed toward individuals with diverse backgrounds and experiences pertinent to the study’s focus [12]. Moreover, snowball and criterion sampling techniques were employed to broaden the participant pool [13].

### 2.1. Sampling

Snowball sampling was used to leverage initial participants to refer others who fit specific criteria and were interested in joining the study, enhancing access to a broader range of individuals beyond traditional recruitment methods [14]. Criterion sampling ensured participants matched particular characteristics vital to the research goals, like SES and dietary habits, facilitating a deeper examination of micronutrient access through the SEM and the impact of socioeconomic elements [15]. Recruitment materials, including flyers, emails, and social media posts, were carefully crafted to outline the study’s aims, methods, and ethical considerations, highlighting the voluntary nature of involvement and the confidentiality of the data to encourage diverse participation [16]. Before joining, potential participants were thoroughly briefed on the study’s details, ensuring informed consent was obtained, which underscored their right to withdraw anytime, aligning with ethical research standards [17]. These strategies underscored the commitment to gathering a diverse and representative sample, crucial for a nuanced analysis of the socioeconomic variables affecting micronutrient availability while maintaining ethical norms of consent and privacy.

### 2.2. Inclusion and Exclusion Criteria

Participants in the study were adults aged 18 and older from various socioeconomic and cultural backgrounds. The inclusion criteria included individuals across different income levels, education statuses, and employment types to investigate factors affecting micronutrient access [18]. All participants had concerns about micronutrient access and nutritional health. The study included participants from different ethnicities and cultural heritages to ensure diverse views [19]. Gender diversity was also considered, with male and female participants included to explore gender-based differences in micronutrient access [20]. Individuals engaged in specific activities or jobs that might impact nutrient access, such as athletes or those in physically demanding roles, were also included [21].

The exclusion criteria aimed to maintain methodological integrity and ethical standards. Participants under 18 were excluded due to legal restrictions on consent without parental involvement [22]. Individuals with cognitive impairments or language barriers that could hinder understanding of the study’s procedures were also excluded to ensure ethical participation [23]. Those in remote locations facing communication or transport issues were excluded to avoid logistical problems and ensure data reliability. Eighteen participants with medical conditions affecting diet or nutrient absorption were excluded to reduce confounding variables [24].

### 2.3. Survey Development

Informed by previous research on cultural perspectives on injury reporting [1,2,3,4,6,7,8], the research team, comprised of two certified athletic trainers (ATs) and two exercise physiologists trained in concussion management and qualitative methods, collaboratively developed an interview protocol. Subsequently, a sports-medicine orthopedic surgeon who is an expert in qualitative research and concussion management reviewed the interview guide to assess the content, completeness, and absence of leading questions. Prior to data collection, the interview protocol underwent pilot testing utilizing three individuals meeting the study’s inclusion criteria and possessing experience with cultural perspectives on injury reporting and concussion management. Consensus was reached among the team, resulting in the finalization of the interview guide.

### 2.4. Data Collection

Data collection for this study had two phases. In the first phase, participants received a letter of solicitation (LOS) with a link to a pre-screening survey on Qualtrics™, where they could confirm their eligibility against the inclusion criteria [25]. The LOS outlined these criteria, explaining what was required [26]. Those who qualified and were interested in participating submitted their email addresses to receive further instructions and start the informed consent process [27].

After screening the responses, the research team contacted individuals who met the inclusion criteria for the next steps, including scheduling interviews [28]. Out of 5000 potential participants contacted through various recruitment methods, 237 started the pre-screening survey, with 198 completing it. Ultimately, 30 participants provided contact information for interview scheduling [29]. This approach ensured that only eligible individuals with relevant experiences and viewpoints were chosen for the study [30].

The study aimed to explore the complex relationships between socioeconomic factors and micronutrient access, focusing on disparities in nutrition [31]. The objective was to understand these dynamics in depth, providing insights to guide policy, interventions, and practices in public health [32]. While qualitative generalization is approached cautiously, this study intended to offer a thorough and detailed thematic analysis, emphasizing specificity rather than broad applicability [19]. This precision is key to rigorous qualitative research [33,34].

To prepare for the data collection phase, the interviewers completed a training session covering the study’s objectives, interview protocol, and ethical considerations. The semi-structured interview format included 12 main questions and additional probing questions to encourage deeper responses (Table 2) [34]. Interviews lasted between 30 and 60 min, and all sessions were recorded and transcribed for analysis (Figure 1) [35].

At the start of each interview, the researchers briefly summarized the study’s aims to remind participants and then reviewed the informed consent form. Any questions from participants were answered for clarity. After addressing their queries, participants were asked to sign and return the consent form electronically. Once consent was given, audio recording began to capture the entire interview session [36].

The interviews followed a structured format with open-ended questions designed to explore the central research question. Researchers used probing and follow-up questions to gather more details when participants’ responses were brief or needed further clarification [37]. The aim was to obtain comprehensive narratives during the interviews, which is vital for qualitative analysis [36]. After the main questions, the Principal Investigator (PI) encouraged participants to share additional thoughts or clarify any unresolved issues, ensuring a thorough discussion [38].

After the interviews, the research team proposed sending the interview transcript via email for review and amendment, and they verified the participants’ contact details for further communication related to the study. The interviews concluded with the team thanking participants for their contribution. The data collection process was systematically organized to maintain consistency and reliability across all interviews [39].

To ensure inter-rater reliability, interviewers independently coded a subset of interviews and then compared and discussed the coding. Regular meetings allowed the interviewers to align their understanding of the responses, maintaining a high level of agreement throughout the data collection process (Figure 2).

### 2.5. Data Analysis

The data analysis, grounded in the SEM, began with transcribing interviews verbatim to ensure accuracy and a thorough understanding of the data (Figure 3) [18]. Instead of using computer-assisted software, the research team manually coded the transcriptions to identify preliminary codes and then grouped them into recurring categories to establish themes [40]. To enhance validity and reliability, three external experts reviewed a sample of the interviews and coding, offering feedback that refined the analysis process [41]. The iterative coding and theme development continued until a consensus was reached, confirming that the themes captured the essence of the data and addressed the research questions [42].

Using the SEM framework, the analysis examined the complex interaction between socioeconomic factors and micronutrient access at various levels of influence. The SEM guided the categorization of data into individual, interpersonal, community, and societal levels, allowing for a detailed examination of how socioeconomic elements and micronutrient availability intersect across different tiers [43]. This approach provided a comprehensive perspective on the factors impacting participants’ access to nutrients, revealing intricate relationships within the data [44].

The coding and theme development process reflected the SEM’s layers, uncovering dynamics like personal economic status and nutritional understanding at the individual level, social networks at the interpersonal level, local area characteristics at the community level, and economic policies at the societal level [45]. This multi-level approach provided a broader view of factors affecting micronutrient access.

Various strategies were employed to ensure the study’s credibility. Peer debriefing involved external feedback on the research process to validate the methodology, strengthening rigor [46]. Member checking confirmed the accuracy of findings with participants, ensuring the analysis accurately reflected their experiences [41]. Detailed audit trails were used to enhance transparency, and confirmability was achieved by ensuring credibility, transferability, and dependability, minimizing researcher bias [47,48]. These measures ensured the findings were robust and representative of participants’ perspectives.

## 3. Results

The study recruited 30 participants for interviews and focus groups to explore various demographic characteristics. Participants were divided by age: 10 (33.3%) were 18–30 years old, 12 (40%) were 31–50 years old, 7 (23.3%) were 51–70 years old, and 1 (3.3%) was over 70 years old. The gender distribution included 13 males (43.3%) and 17 females (56.7%). Regarding ethnicity, 60% identified as White/Caucasian, 20% as Black/African American, 10% as Hispanic/Latino, and 10% as Asian. Regarding educational levels, 16.7% had a high school education or less, 33.3% had some college or an associate’s degree, 26.7% held a bachelor’s degree, and 23.3% had a graduate or professional degree. These demographics offer a comprehensive examination of the socioeconomic factors influencing micronutrient access.

Socioeconomic status (SES) and geographic location were also examined. Among the 30 participants, 50% had low household incomes, 33.3% were in the middle-income category, and 16.7% were in the high-income bracket. Regarding employment, 66.7% were employed full-time, 16.7% were part-time, and 16.7% were unemployed. Food security data showed that 60% experienced some level of food insecurity in the past year. For geographic location, 66.7% lived in urban areas, while the rest were in rural or suburban settings. These factors give further insight into participants’ diversity and socioeconomic conditions.

In addition, the study considered participants’ primary language spoken at home and their native language. Of the 30 participants, 66.7% primarily spoke English at home, 16.7% spoke Spanish, 10% spoke Mandarin, and 6.7% spoke Urdu. Regarding native language, 83.3% identified English, 10% identified Spanish, and 6.7% identified Mandarin. All participants were fluent in English. These linguistic demographics provide additional context for the study, indicating how language and cultural backgrounds intersect with socioeconomic factors to affect micronutrient access and dietary behaviors.

### 3.1. Themes

Table 3 illustrates the five overarching themes identified through the data analysis process to investigate the research question.

**Theme** **1.**
*Individual-Level Factors—This theme encompasses personal behaviors, nutritional knowledge, and dietary preferences.*


Theme 1, Individual-Level Factors, delves into participants’ personal behaviors, nutritional knowledge, and dietary preferences. It explores how individual characteristics and choices influence access to and the consumption of essential micronutrient-rich foods. Understanding these factors provides insight into the internal determinants that shape dietary behaviors and nutritional outcomes (Table 4).

Theme 1, Individual-Level Factors, sheds light on the intricate interplay between personal behaviors, nutritional knowledge, and dietary preferences in shaping individuals’ access to essential micronutrients. Participants provided valuable insights into their personal dietary habits, highlighting the role of routine and preference in their food choices. For instance, one participant (P7) remarked, “I try to stick to a balanced diet, but I have a weakness for sweets. I know it’s not the healthiest, but it’s hard to resist sometimes”. This quote reflects individuals’ challenges in balancing nutritional goals with personal indulgences.

Moreover, participants demonstrated varying levels of nutritional awareness, with some expressing confidence in their knowledge of healthy eating practices. Another participant (P12) articulated, “I ensure to include plenty of fruits and veggies in my meals. I know they’re packed with vitamins and minerals that my body needs”. This quote underscores the importance of nutritional education in guiding dietary choices and promoting micronutrient-rich diets.

Participants also shared insights into their food preferences and how they influence their dietary behaviors. “I grew up eating traditional dishes from my culture, and I still prefer those foods over anything else”, shared one participant (P5). This sentiment highlights the cultural and personal factors that shape individuals’ food choices and dietary patterns.

Theme 1 elucidates the multifaceted nature of individual-level factors in determining access to and the consumption of essential micronutrients. Through participants’ narratives, we understand the diverse factors that inform dietary behaviors and underscore the importance of personalized approaches to nutrition education and intervention strategies.

**Theme** **2.**
*Interpersonal Relationships—This theme focuses on the role of family, friends, and social networks in influencing dietary choices.*


Theme 2, Interpersonal Relationships, reflects the diverse aspects of interpersonal influences on participants’ dietary behaviors and preferences. “Family Traditions” encapsulates the impact of cultural practices and familial customs on participants’ food choices, as evidenced by 35 transcript excerpts contributed by 15 participants. “Peer Influence” highlights the role of friends and social circles in shaping dietary decisions, with 28 transcript excerpts from 12 participants illustrating the impact of peer pressure and social norms. “Social Gatherings” encompasses discussions surrounding food-related events and social occasions, with 22 transcript excerpts from 10 participants shedding light on the influence of social environments on eating behaviors. Lastly, “Social Support” reflects the supportive role of social networks in promoting healthier eating habits, with 40 transcript excerpts from 17 participants emphasizing the importance of social connections in fostering dietary changes. These codes and their respective frequencies underscore the significance of interpersonal relationships in shaping individuals’ dietary behaviors, highlighting the need to consider social influences in nutrition interventions and health promotion efforts (Table 5).

Theme 2, Interpersonal Relationships, delves into the significance of family, friends, and social networks in shaping individuals’ dietary choices and behaviors. Participants highlighted the influential role of their immediate social circles in terms of their food preferences and eating habits. For instance, many participants mentioned how family traditions and cultural practices influenced their dietary choices. One participant (P9) expressed, “My family has always emphasized home-cooked meals and shared dinners. It’s not just about the food; it’s about bonding and connecting with each other over a meal”. This sentiment underscores food consumption’s social and cultural dimensions within familial contexts.

Moreover, participants discussed the impact of peer influences and social networks on their dietary behaviors. Some participants noted how social gatherings and events often revolve around food, influencing their choices and consumption patterns. As articulated by another participant (P14), “When I go out with friends, it’s hard to resist ordering unhealthy foods. There’s this social pressure to indulge, even if it’s not what I would normally eat”. This quote highlights the social dynamics in shaping individuals’ dietary decisions and underscores the need to consider social influences in nutrition interventions.

Furthermore, participants shared experiences of receiving support and encouragement from their social networks to adopt healthier eating habits. “My friends and I started a cooking club where we share healthy recipes and meal ideas. It’s been great having that support system”, mentioned one participant (P3). This example illustrates how social support can positively impact individuals’ dietary behaviors and promote healthier eating patterns.

Theme 2 elucidates the intricate interplay between interpersonal relationships and dietary choices, highlighting the importance of familial, peer, and social influences in shaping individuals’ food preferences and consumption behaviors. Through participants’ narratives, we gain insights into how social networks impact nutritional decisions and underscore the need to leverage social support to promote healthy eating habits.

**Theme** **3.**
*Community Environment—This theme explores access to food stores, community programs, local policies, and neighborhood environments.*


Theme 3, Community Environment, delves into the influence of external factors such as access to food stores, community programs, local policies, and neighborhood environments on individuals’ dietary choices and behaviors. This theme sheds light on how communities’ physical and social aspects shape food availability, affordability, and quality, influencing individuals’ access to essential nutrients. Through participants’ insights, this theme elucidates the multifaceted relationship between community environments and dietary behaviors, highlighting the importance of addressing environmental determinants to promote healthier eating habits and improve nutritional outcomes (Table 6).

Table 6 presents the codes that emerged while analyzing Theme 3: Community Environment. These codes represent various aspects of the community environment that influence individuals’ access to nutritious foods and dietary behaviors. “Food Accessibility” reflects participants’ discussions on the availability and proximity of food stores and markets, with 42 transcript excerpts contributed by 18 participants highlighting challenges and disparities in accessing healthy food options. “Community Resources” encompass conversations regarding the presence of community programs and initiatives aimed at promoting nutrition and food security, with 30 transcript excerpts from 14 participants illustrating the role of community resources in facilitating access to nutritious foods. “Local Policies” delineates participants’ perceptions of government regulations and policies related to food and nutrition, with 20 transcript excerpts from 9 participants discussing the impact of policy interventions on food environments. Lastly, “Neighborhood Characteristics” captures discussions on the social and physical attributes of neighborhoods that influence dietary behaviors, with 35 transcript excerpts from 16 participants highlighting the significance of neighborhood factors in shaping food choices and consumption patterns. These codes illuminate the complex interplay between community environments and dietary behaviors, underscoring the need for targeted interventions to address environmental determinants and promote equitable access to nutritious foods within communities.

**Theme** **4.**
*Societal Factors: Examines the impact of economic policies, national nutrition programs, societal norms, and public health initiatives.*


Theme 4, Societal Factors, explores the broader societal influences on individuals’ dietary behaviors, including economic policies, national nutrition programs, societal norms, and public health initiatives. This theme delves into how macro-level factors shape individuals’ access to and consumption of essential nutrients, highlighting the systemic barriers and facilitators that impact nutritional outcomes. Through participants’ perspectives, Theme 4 elucidates the interplay between societal structures and dietary behaviors, emphasizing the need for comprehensive strategies and policy interventions to address socioeconomic disparities and promote equitable access to healthy food options (Table 7).

Table 7 illustrates the codes derived from Theme 4: Societal Factors, reflecting the diverse societal influences on individuals’ dietary behaviors and nutritional outcomes. For instance, under “Economic Policies”, participants voiced concerns about the impact of economic disparities on food access and affordability. One participant (P7) remarked, “The rising cost of healthy foods makes it difficult for low-income families to maintain a nutritious diet”. This sentiment was echoed by others, contributing to 28 transcript excerpts from 12 participants highlighting the role of economic policies in shaping dietary choices.

“National Nutrition Programs” emerged from discussions surrounding government initiatives aimed at enhancing nutrition and food security. A participant (P10) shared, “Programs like SNAP and WIC are vital for families struggling to put food on the table”. This sentiment resonated with others, resulting in 18 transcript excerpts from 8 participants discussing the accessibility and effectiveness of national nutrition programs.

“Societal Norms” encapsulated participants’ perceptions of cultural beliefs and societal expectations regarding food and nutrition. One participant (P4) remarked, “In our culture, there’s this notion that eating fast food is more convenient than cooking at home”. Such cultural norms influenced dietary behaviors, contributing to 24 transcript excerpts from 10 participants.

Lastly, “Public Health Initiatives” captured discussions on the role of health campaigns and interventions in promoting healthy eating habits. A participant (P12) stated, “Health education programs in schools have raised awareness about the importance of eating fruits and vegetables”. This sentiment, along with others, led to 32 transcript excerpts from 14 participants illustrating the impact of public health initiatives on dietary behaviors. These quotes exemplify how societal factors intersect with individuals’ dietary choices, highlighting the need to address broader socioeconomic determinants to foster environments conducive to healthy eating habits and improved nutritional outcomes.

**Theme** **5.**
*Intersectionality: Considers the complex interactions between multiple levels of influence, highlighting the interconnectedness of individual, interpersonal, community, and societal factors.*


Theme 5, Intersectionality, delves into the intricate interplay between various levels of influence, emphasizing the interconnectedness of individual, interpersonal, community, and societal factors in shaping dietary behaviors and nutritional outcomes. This theme recognizes that individuals’ experiences are shaped by many intersecting identities and circumstances, leading to unique challenges and opportunities in accessing and incorporating essential nutrients into their diets (Table 8).

Table 8 presents the codes that emerged while analyzing Theme 5: Intersectionality. These codes highlight the complex interactions between various levels of influence, shedding light on the nuanced ways in which intersecting identities, structural barriers, cultural influences, and environmental contexts shape individuals’ dietary behaviors and nutritional outcomes.

“Intersecting Identities” reflects participants’ discussions on how multiple aspects of their identity, such as race, gender, SES, and age, intersect to influence their dietary choices. One participant (P5) remarked, “As a low-income person of color, I often face challenges accessing fresh produce in my neighborhood”. This sentiment was echoed by others, contributing to 35 transcript excerpts from 15 participants.

“Structural Barriers” encompasses conversations regarding systemic obstacles that impede individuals’ access to nutritious foods, such as food deserts, transportation limitations, and economic inequalities. A participant (P8) shared, “Living in a food desert makes it hard to find affordable fruits and vegetables nearby”. This sentiment resonated with others, resulting in 28 transcript excerpts from 12 participants discussing the impact of structural barriers on dietary choices.

“Cultural Influences” delineates participants’ perceptions of cultural norms, traditions, and practices related to food and nutrition. One participant (P3) remarked, “In my culture, food is a central part of social gatherings, and we often indulge in rich, high-calorie dishes”. Such cultural influences shape dietary behaviors, contributing to 22 transcript excerpts from 10 participants.

“Environmental Context” captures discussions on the physical and social environments in which individuals make food choices. A participant (P11) stated, “Living in a neighborhood with limited grocery options makes it challenging to follow a healthy diet”. This sentiment and others led to 30 transcript excerpts from 14 participants illustrating the impact of environmental context on dietary behaviors.

These codes underscore the importance of considering intersectionality in understanding individuals’ dietary behaviors. They highlight the need for interventions and policies that address the complex interplay of intersecting identities and structural factors to promote equitable access to essential nutrients and improve nutritional outcomes for all individuals.

### 3.2. Results Summary

The results of this qualitative study reveal the complex determinants of micronutrient access and consumption, highlighting the interplay of individual, interpersonal, community, and societal factors. Using thematic analysis guided by the SEM, five major themes emerged, shedding light on influences that shape dietary behaviors and nutritional outcomes.

At the individual level, personal behaviors, nutritional knowledge, and dietary preferences were key in determining food choices. Interpersonal factors, including family, friends, and social networks, played a significant role in influencing participants’ dietary decisions, illustrating the impact of social support on healthy eating. Community environments, like access to food stores and local policies, affected food availability, with disparities in neighborhood settings contributing to different dietary outcomes.

Societal factors, such as economic policies, national nutrition programs, societal norms, and public health initiatives, also influenced participants’ dietary behaviors. Additionally, the intersectionality of multiple levels emphasized the need for tailored interventions to address intersecting identities and structural barriers to ensure equitable access to essential nutrients.

These results suggest a need for holistic strategies that integrate individual, interpersonal, community, and societal dimensions to improve nutritional outcomes and promote health equity across different groups. Appendix A contains selected participant quotes to support the thematic analysis, reflecting their experiences and supporting the study’s findings. These quotes directly connect participant narratives and the thematic categories, adding credibility to the analysis and providing deeper insights into the thematic framework.

## 4. Discussion

In this study, we explored micronutrient access through the SEM, utilizing a diverse range of demographic variables such as age, gender, ethnicity, educational levels, and socioeconomic status to deepen our understanding of how various factors influence dietary behaviors and micronutrient availability. Collecting detailed demographic information was crucial, enabling the analysis of how different socioeconomic and cultural factors intersect to impact health outcomes. For example, variations in age highlighted evolving nutritional needs and access issues across the life course. At the same time, differences in educational levels provided insights into the correlation between nutrition knowledge and dietary practices. This comprehensive demographic analysis adheres to the SEM principles, emphasizing the interaction between multiple influence levels and enriching the study’s findings, which is essential for designing targeted public health interventions.

### 4.1. Theme 1: Individual-Level Factors

Several key findings emerged in Theme 1, which explored individual-level factors related to dietary behaviors and preferences. Participants discussed various aspects of their personal behaviors, nutritional knowledge, and dietary preferences, shedding light on the complexity of factors influencing their food choices. The analysis revealed that various factors shaped participants’ dietary habits, including personal preferences, health-conscious choices, and dietary restrictions [49]. Participants described making health-conscious choices based on their nutritional awareness and dietary goals, highlighting the importance of individual agency in shaping dietary behaviors. Additionally, factors such as meal planning, cooking skills, and food budgeting emerged as influential in determining participants’ access to and consumption of nutritious foods.

Interpreting these findings within the context of the SEM theoretical framework, it becomes evident that individual-level factors interact with broader environmental and societal influences to shape dietary behaviors [10]. The SEM posits that individual behaviors are influenced by factors at multiple levels, including interpersonal relationships, community environments, and societal norms. In the context of this study, the findings suggest that while individual-level factors play a significant role in shaping dietary behaviors, they are also influenced by external factors such as access to food resources, SES, and cultural norms.

The significance of these findings lies in their implications for interventions aimed at promoting healthy eating habits and improving nutritional outcomes. By understanding the interplay between individual-level factors and broader environmental influences, policymakers and public health practitioners can develop targeted strategies to address barriers to healthy eating at multiple levels. For example, interventions focused solely on individual-level factors may overlook the structural barriers that limit access to nutritious foods for marginalized communities. Instead, a comprehensive approach that addresses both individual behaviors and environmental determinants is needed to create environments that support healthy food choices for all individuals.

In comparing the findings of Theme 1 with the existing literature, several notable similarities and differences emerge. Existing research in the field of nutrition and dietary behavior has highlighted the importance of individual-level factors, such as personal preferences and dietary habits, in shaping food choices [49]. Consistent with previous studies, our findings underscored the significance of factors like health-conscious choices, dietary restrictions, and nutritional awareness in influencing participants’ dietary behaviors [50].

However, our study also revealed unique insights into the nuanced interplay between individual-level factors and broader environmental influences, which may not have been fully captured in previous research. For example, while existing literature has acknowledged the role of personal behaviors in dietary decision-making, our study elucidated how these behaviors are influenced by external factors such as access to food resources, cooking skills, and food budgeting [51].

By contextualizing our findings within the SEM theoretical framework, we can further elucidate the significance of these insights. The SEM posits that individual behaviors are shaped by factors at multiple levels of influence, including interpersonal relationships, community environments, and societal norms [10]. Our study contributes to the existing body of knowledge by comprehensively understanding how individual-level factors interact with broader environmental influences to shape dietary behaviors.

Our findings align with previous research on individual-level factors in dietary behavior while offering novel insights into the complex interplay between individual behaviors and environmental determinants. By identifying similarities and differences with the existing literature, our study enriches the field’s understanding of the factors influencing food choices. It underscores the importance of considering multiple levels of influence in interventions aimed at promoting healthy eating habits.

### 4.2. Theme 2: Interpersonal Relationships

Theme 2 delves into the role of interpersonal relationships, specifically focusing on the influence of family, friends, and social networks on dietary choices. The data revealed that participants often relied on social connections for support, guidance, and encouragement regarding their food decisions. Interpersonal relationships emerged as significant determinants of dietary behaviors, with participants citing the impact of family traditions, peer influences, and social support networks on their food choices. These findings underscore the importance of considering interpersonal dynamics in understanding individuals’ dietary behaviors and nutritional outcomes.

Utilizing the SEM theoretical framework, our interpretation of these findings emphasizes the interconnectedness between individual-level factors and interpersonal relationships. The SEM posits that interpersonal relationships play a crucial role in shaping individuals’ behaviors and beliefs, with social networks serving as channels for information exchange and social influence [10]. Our study contributes to this framework by highlighting how interpersonal relationships influence dietary behaviors, reflecting the dynamic interplay between individual agency and social context.

Moreover, the significance of interpersonal relationships in dietary decision-making aligns with previous research emphasizing the role of social support and social norms in shaping health behaviors [52]. By examining the influence of family, friends, and social networks on dietary choices, our study adds to the existing literature on social determinants of health. It provides insights into how interpersonal relationships impact nutritional outcomes.

The findings of Theme 2 underscore the importance of considering interpersonal dynamics in understanding individuals’ dietary behaviors. By elucidating the role of family, friends, and social networks in shaping food choices, our study contributes to a comprehensive understanding of the factors influencing nutritional outcomes. It highlights the need for interventions that address social influences on dietary behaviors.

In comparing our findings on interpersonal relationships with the existing literature, we note several similarities and differences that contribute to the broader understanding of dietary behaviors within the context of social influences. Previous research has consistently highlighted the significant role of social networks, family dynamics, and peer influences in shaping individuals’ dietary choices [6]. Our study aligns with these findings, emphasizing the importance of interpersonal relationships as key determinants of food decisions.

However, our study also offers unique insights into the specific mechanisms through which interpersonal relationships influence dietary behaviors. For instance, we found that family traditions and cultural practices played a central role in shaping participants’ food choices, highlighting the cultural embeddedness of dietary behaviors within social networks. This nuanced understanding adds depth to the existing literature by elucidating the complex interplay between cultural norms, social support, and dietary preferences.

Furthermore, our study extends previous research by examining the role of social networks beyond immediate family members to include broader social circles, such as friends and colleagues. We found that peer influences and social support networks outside the family unit also exerted considerable influence on participants’ food decisions, echoing the findings of studies emphasizing the impact of peer relationships on health behaviors [53].

Utilizing the SEM theoretical framework, our study contributes to theoretical advancements in understanding the social determinants of dietary behaviors. Our findings underscore the dynamic interactions between individual agency, interpersonal dynamics, and broader social structures in shaping food choices by situating interpersonal relationships within the broader socio-ecological context [10].

In conclusion, our study adds to the existing body of knowledge by providing nuanced insights into the role of interpersonal relationships in influencing dietary behaviors. By considering the complex interplay between family traditions, peer influences, and social support networks, our findings contribute to a comprehensive understanding of the social determinants of nutrition and highlight avenues for future research and intervention.

### 4.3. Theme 3: Community Environment

The investigation into Theme 3, “Community Environment”, elucidates the profound influence of community infrastructure on individual dietary patterns and micronutrient access. This theme underscores the intertwined roles of food accessibility, community resources, local policies, and neighborhood characteristics in shaping nutritional outcomes.

Food Accessibility is paramount, as demonstrated by 42 transcript excerpts from 18 participants. The accessibility and proximity of food stores are crucial determinants of dietary choices and micronutrient consumption. This aligns with research by Walker, Keane, and Burke (2010), which found that the availability of supermarkets and grocery stores near residential areas significantly correlates with improved fruit and vegetable intake and better overall dietary quality [54]. The challenges highlighted by participants in accessing affordable, nutrient-rich foods underscore the critical need for strategically located food outlets.

Community resources play a vital role in nutritional well-being, as evidenced by 30 excerpts from 14 participants. Community-led initiatives like gardens, educational programs, and subsidized food schemes are essential in promoting food security and healthy eating habits. A study by Freedman, Blake, and Liese (2013) supports this, indicating that community gardens and educational programs can enhance fruit and vegetable consumption and improve food literacy among community members [55]. These resources are pivotal in bridging the gap in food access, especially in underserved areas.

Local policies have a significant impact on food access and affordability, highlighted in 20 excerpts from 9 participants. Governmental policies and regulations can either facilitate or obstruct access to nutritious food, emphasizing the importance of policy alignment with public health goals. According to Morland and Filomena (2007), zoning laws and food retail licensing can influence the density of food outlets and the availability of healthy food options in a community [56]. This evidence underscores the necessity for policy interventions that prioritize establishing food environments conducive to healthy eating.

The 35 excerpts from 16 participants discuss neighborhood characteristics, which reveal the socioeconomic and physical dimensions of neighborhoods that affect food choices and availability. Studies such as those by Zenk et al. (2005) have illustrated how neighborhood SES and urban–rural distinctions can create disparities in access to healthy foods [57]. The neighborhood environment, including the prevalence of food deserts and economic disparities, plays a crucial role in dietary behaviors and micronutrient intake.

In synthesizing the findings of Theme 3 with the existing literature, it becomes evident that the community environment significantly influences dietary choices and micronutrient access. The complex interplay between food accessibility, community resources, local policies, and neighborhood characteristics necessitates comprehensive, multi-level interventions. To enhance micronutrient access and promote healthier dietary behaviors, it is imperative to implement strategies that address these community-level determinants. These include improving the availability of affordable nutritious foods, strengthening community nutritional programs, enacting supportive local policies, and tailoring interventions to the unique characteristics of each neighborhood. By aligning community resources and policies with public health objectives, we can create supportive environments that facilitate better dietary choices and improve nutritional outcomes across diverse populations.

### 4.4. Theme 4: Societal Factors

Theme 4, “Societal Factors”, investigates the extensive impact of economic policies, national nutrition programs, societal norms, and public health initiatives on individuals’ dietary choices and micronutrient access. This theme is pivotal in understanding the broader systemic influences shaping nutritional outcomes at the societal level.

Economic Policies: With 28 excerpts from 12 participants, this code highlighted the profound influence of economic policies on food affordability and availability. Economic strategies, such as taxation on sugary foods and subsidies for healthy options, directly affect the cost of food and, consequently, individuals’ ability to purchase nutrient-rich foods. Studies like that of Thow, Jan, Leeder, and Swinburn (2010) have shown how fiscal policies can influence dietary behaviors by making healthier food options more financially accessible or deterring unhealthy food consumption through higher taxes [58].

National Nutrition Programs: Discussed in 18 excerpts from 8 participants, this code underlines the critical role of programs like SNAP and WIC in providing food security and nutritional support to low-income families. Research by Andreyeva, Tripp, and Schwartz (2015) demonstrates these programs’ effectiveness in improving participants’ dietary quality and enhancing their micronutrient intake and overall health [59].

Societal Norms: This code, represented in 24 excerpts from 10 participants, sheds light on the cultural and societal expectations that influence dietary habits. The impact of societal norms on food choices is significant, as they dictate the perceived desirability and acceptance of certain foods within communities. The work of Pachucki, Jacques, and Christakis (2011) illustrates how social and cultural norms can shape food preferences and eating patterns, often leading to disparities in diet quality across different socioeconomic groups [60].

Public Health Initiatives: With 32 excerpts from 14 participants, this code emphasizes the importance of initiatives aimed at promoting healthy eating and improving public health. Effective public health campaigns and programs can significantly influence population-wide dietary behaviors, as indicated by the success of initiatives like the “5 A Day” campaign in increasing fruit and vegetable intake among the general public, as documented by John, Ziebland, Yudkin, Roe, and Neil (2002) [61].

The interplay of these societal factors illustrates a multi-layered approach to addressing the challenges of micronutrient access and dietary behavior. Economic policies, national nutrition programs, societal norms, and public health initiatives are all integral in shaping the food environment and influencing individuals’ nutritional choices. A holistic approach encompassing these factors is essential to formulate effective strategies for improving dietary quality and ensuring equitable access to micronutrients across different populations.

### 4.5. Theme 5: Intersectionality

Theme 5, “Intersectionality”, examines the nuanced interplay of individual, interpersonal, community, and societal factors. It reveals how these layered influences collectively shape dietary behaviors and micronutrient access. This theme emphasizes the multifaceted nature of nutritional health, where various determinants interact in complex ways.

Intersecting Identities: This code, discussed in 35 excerpts from 15 participants, underscores the ways in which multiple aspects of identity—such as race, gender, SES, and age—converge to affect dietary choices and nutrition. The concept of intersecting identities is supported by the work of Hankivsky and Cormier (2009), who emphasize the importance of considering multiple and overlapping social factors when analyzing health outcomes [62]. People’s experiences and opportunities, including their access to nutritious foods, are profoundly shaped by their identities and the way these identities intersect.

Structural Barriers: Highlighted in 28 excerpts from 12 participants, this code refers to the systemic obstacles that impede access to healthy foods, such as food deserts, economic inequality, and inadequate transportation. Studies like that by Larson et al. (2009) illustrate how structural barriers, including geographic and economic factors, limit individuals’ ability to obtain nutritious food, thereby affecting their dietary habits and health [7].

Cultural Influences: With 22 excerpts from 10 participants, this code reflects the impact of cultural norms, traditions, and practices on food choices and dietary behaviors. The influence of culture on nutrition has been explored by researchers like Airhihenbuwa et al. (1996), who argue that cultural beliefs and values significantly shape individuals’ food preferences and dietary practices, often leading to varied nutritional outcomes across different cultural groups [63].

Environmental Context: Discussed in 30 excerpts from 14 participants, this code considers the role of the physical and social environment in influencing dietary decisions. The work of Cummins and Macintyre (2006) supports this, demonstrating that the availability of food resources in the local environment, including grocery stores and food markets, plays a crucial role in shaping individuals’ dietary choices and nutritional status [64].

The intersectionality theme demonstrates that dietary behaviors and micronutrient access result from dynamic interaction among various factors at different levels of influence. Understanding these complex relationships is vital for developing effective nutritional interventions that are sensitive to the diverse needs and circumstances of different population groups. A holistic approach that addresses the multifactorial nature of dietary choices and nutrition is essential to promote equitable access to healthy foods and improve public health outcomes.

### 4.6. Revisiting the Social Ecological Model: Insights and Implications from a Nutritional Perspective

This study’s examination of micronutrient access and dietary behaviors adds depth to the SEM, affirming its multi-layered approach while exposing some challenges and extensions to its traditional framework. The findings validate the SEM’s ability to illustrate the complex factors impacting nutritional health by breaking down individual, interpersonal, community, and societal influences. The results support the SEM’s concept that health behaviors are not merely individual choices but result from various environmental and social interactions.

The study also challenges the SEM by showing that the traditional framework might not capture the full complexity of these interactions. The intersectionality theme, for example, demonstrates how different SEM levels are interconnected, with one level’s effects often depending on factors at another level. This indicates that the SEM’s common representation as concentric circles may oversimplify real-world interactions.

Additionally, the research extends the SEM by exploring how these layers interact over time and in varying contexts. It introduces a more dynamic perspective, suggesting that influences on health behavior change as policies, societal norms, and economic conditions evolve. This dynamic view implies that public health strategies must be flexible and responsive to the shifting nature of these determinants.

This study not only supports the foundational concepts of the SEM but also pushes its theoretical boundaries by highlighting the model’s complexities and dynamism. These insights encourage a rethinking of how the SEM is used in public health and nutrition research, promoting more holistic and adaptable approaches to address the multiple factors influencing dietary behavior and micronutrient access.

### 4.7. Implications for Theory Development

The broader implications of the study’s findings extend across theoretical development, policy formulation, practice enhancement, and future research directions, addressing significant gaps in the nutritional and public health literature. The study provides a comprehensive examination of the SEM in the context of nutrition, highlighting the intricate and dynamic interplay of factors influencing dietary behavior and micronutrient access. It challenges and extends the SEM by emphasizing the importance of intersectionality and the complex interactions between individual, interpersonal, community, and societal factors. This nuanced understanding prompts a reevaluation of existing health behavior theories, advocating for models that incorporate the fluid and interconnected nature of these influences. The findings encourage a more integrative theoretical approach that acknowledges the compounded effects of various determinants on health outcomes.

### 4.8. Implications for Policy

From a policy perspective, this study underscores the necessity of multi-level interventions to address nutritional inequalities. The evidence points to the need for comprehensive policies that go beyond individual behavior change and target the broader socioeconomic and environmental structures. For instance, the findings related to economic policies and structural barriers suggest that initiatives should promote healthy food choices and enhance access and affordability of nutritious foods through subsidies, zoning laws, and support for local food systems. Policymakers are encouraged to use these insights to develop holistic strategies that address the root causes of dietary disparities and ensure equitable access to healthy foods for all population groups.

### 4.9. Implications for Practice

The study’s findings highlight the importance of tailored interventions considering different communities’ diverse needs and circumstances. Health practitioners and nutritionists are urged to adopt a more contextualized approach to dietary guidance, recognizing the various external factors influencing individuals’ food choices. The insights into community and environmental contexts, for example, suggest that nutrition education and intervention programs should be designed with an awareness of the local availability of food resources and cultural dietary preferences.

### 4.10. Implications for Future Research

The study identifies several avenues for future research. Longitudinal studies are needed to understand how changes in socioeconomic, environmental, and policy contexts over time affect dietary behaviors and nutritional health. Additionally, the concept of intersectionality introduced in this research calls for more studies that examine how overlapping social identities and structural factors influence nutrition, particularly in under-researched populations. The findings also suggest a potential for developing and testing new integrative models that capture the complex and dynamic nature of health behaviors identified through this research.

### 4.11. Addressing Gaps in the Literature

The study addresses practical and theoretical gaps in the literature by thoroughly examining the factors influencing micronutrient access and dietary behavior across different socioeconomic groups. It moves beyond individual-level analyses common in nutritional research, offering a comprehensive view that integrates personal, social, and environmental dimensions. This approach not only fills a gap in understanding the broader determinants of dietary behavior but also offers a more holistic framework for developing interventions to improve nutritional health and reduce disparities.

This research’s broader implications are far-reaching, impacting theory, policy, practice, and future nutrition and public health research. The study paves the way for more effective, equitable, and context-sensitive approaches to promoting nutritional health and well-being by offering new insights into the complex factors affecting dietary behaviors and micronutrient access.

### 4.12. Study Limitations

This study has several limitations that should be considered when interpreting the findings. Although the sample size of 30 participants is adequate for qualitative research and surpasses the typical threshold for data saturation, it may still limit the generalizability of the results to broader populations, as the specific socioeconomic and cultural backgrounds of the participants might influence the themes identified, making the findings more relevant to similar contexts rather than universally applicable. Additionally, the study relies on self-reported data, which can introduce biases such as social desirability or recall bias, where participants may not accurately report their dietary habits. While data saturation was achieved with 20 interviews, the additional 10 interviews were conducted to utilize already scheduled participants, reinforcing saturation, but they may not have added significant new insights. Furthermore, the interpretative nature of qualitative data analysis might introduce subjectivity despite efforts to maintain rigor through methods such as peer debriefing and inter-rater reliability checks. These limitations highlight the need for caution in extending the findings beyond the study setting and suggest areas for further research to confirm and expand upon the results presented.

## 5. Conclusions

This study uses the SEM to explore the relationship between socioeconomic factors and micronutrient access, demonstrating how individual, interpersonal, community, and societal levels influence dietary behaviors and nutritional outcomes. It highlights the roles of social support networks and socioeconomic policies in shaping access to essential micronutrients, suggesting the need for public health initiatives that comprehensively address socioeconomic determinants. Research confirms that interventions must go beyond individual behavior to tackle structural determinants of health, with the introduction of intersectionality providing insights into the challenges faced by diverse populations. The results support the development of targeted interventions and policies aimed at improving micronutrient access and reducing dietary disparities, with future research suggested to focus on longitudinal studies across various socioeconomic settings. By applying the SEM and supporting its relevance with empirical data, this study contributes to the fields of nutrition and public health, aiming to enhance the well-being of populations in different socioeconomic contexts.

## Figures and Tables

**Figure 1 nutrients-16-01757-f001:**
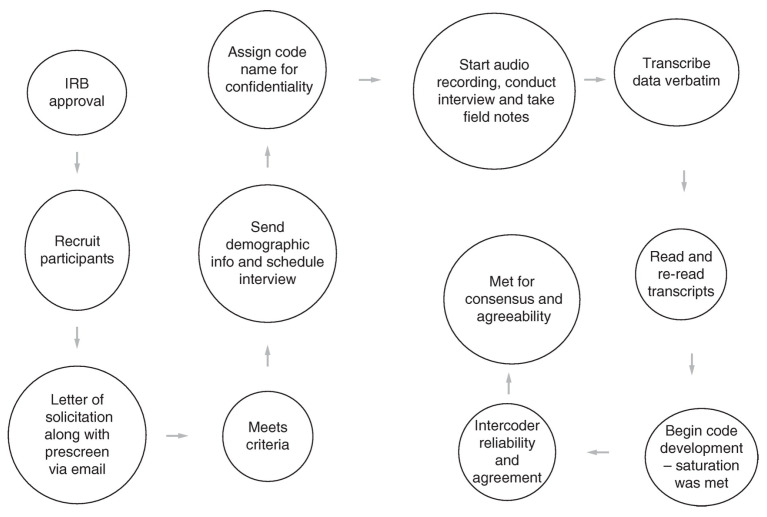
Interview process.

**Figure 2 nutrients-16-01757-f002:**
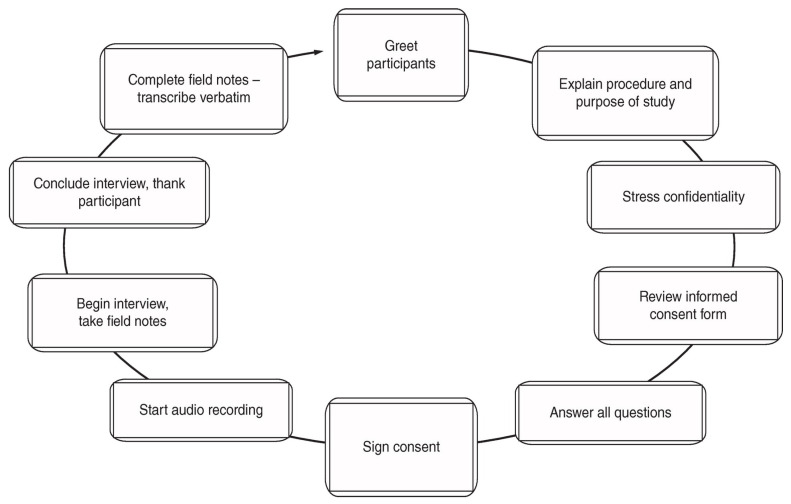
Data collection process.

**Figure 3 nutrients-16-01757-f003:**
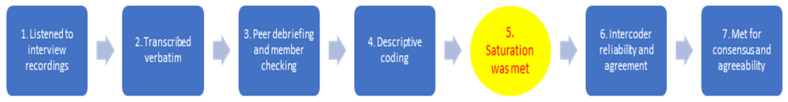
Data analysis process.

**Table 1 nutrients-16-01757-t001:** Social Ecological Model: key factors influencing micronutrient access.

Level of Influence	Key Factors
Individual	· Personal behaviors: Choices regarding food consumption, meal planning, and dietary habits.
	· Nutritional knowledge: Awareness of nutrient-rich foods and their health benefits.
	· Dietary preferences: Taste preferences, cultural considerations, and dietary restrictions influencing food choices.
Interpersonal	· Family: Household food purchasing and meal preparation practices influencing access to nutrient-rich foods.
	· Friends and social networks: Opportunities for sharing food-related information and resources.
	· Social support: Impacting individuals’ ability to obtain essential nutrients through shared resources and support systems.
Community	· Access to food stores: Availability and accessibility of grocery stores, supermarkets, and farmers’ markets.
	· Community programs: Food assistance programs, nutrition education initiatives, and community gardens promoting access to nutrients.
	· Local policies: Zoning regulations, food environments, and built environments affecting the affordability and accessibility of healthy foods.
Societal	· Economic policies: Income inequality, employment opportunities, and food pricing strategies influencing purchasing power.
	· National nutrition programs: Food assistance programs, school meal programs, and nutritional guidelines ensuring equitable access to essential nutrients.
	· Societal norms: Cultural practices surrounding food consumption and dietary patterns influencing food choices.
	· Public health initiatives: Programs promoting healthy eating, addressing food insecurity, and reducing disparities in food access.

**Table 2 nutrients-16-01757-t002:** Interview questions.

Question Number	Interview Question	Probes (If Necessary)
1	Can you describe your typical dietary habits and food choices?	Can you provide specific examples of foods you commonly consume?
2	How do you perceive the affordability of foods rich in essential micronutrients?	Can you describe any challenges you face in accessing affordable nutritious foods?
3	What factors influence your decisions when selecting foods to purchase or consume?	Can you discuss any external influences, such as advertising or societal norms, on your food choices?
4	How do your family and friends impact your dietary behaviors?	Can you provide examples of situations where your social network influenced your food choices?
5	What role do community resources, such as grocery stores or farmers’ markets, play in your access to nutrient-rich foods?	How does the availability of healthy food options in your community influence your dietary decisions?
6	How do economic factors, such as income level or employment status, affect your ability to access and consume nutritious foods?	Can you discuss any strategies you use to manage food expenses while maintaining a healthy diet?
7	Are there any governmental policies or programs that you feel positively or negatively impact your ability to access nutrient-rich foods?	Can you provide examples of how these policies or programs have influenced your dietary choices?
8	How do cultural or societal norms influence your dietary behaviors?	Can you describe any traditional dietary practices or cultural beliefs that influence your food choices?
9	What role does personal knowledge about nutrition play in your food choices?	Can you discuss any experiences or sources that have contributed to your understanding of nutrition?
10	Have you encountered any barriers or challenges in accessing or consuming foods rich in essential micronutrients?	Can you describe specific instances where you struggled to obtain or incorporate nutritious foods into your diet?
11	How do you perceive the support provided by your social network in maintaining a healthy diet?	Can you discuss any instances where your social network has encouraged or discouraged healthy eating habits?
12	In what ways do you think interventions or policies could improve access to nutrient-rich foods in your community?	Can you suggest any specific initiatives or changes that you believe would enhance access to healthy foods in your area?

**Table 3 nutrients-16-01757-t003:** Themes.

Theme Number	Theme Description
1	Individual-Level Factors: This theme encompasses personal behaviors, nutritional knowledge, and dietary preferences.
2	Interpersonal Relationships: Focuses on the role of family, friends, and social networks in influencing dietary choices.
3	Community Environment: Explores access to food stores, community programs, local policies, and neighborhood environments.
4	Societal Factors: Examines the impact of economic policies, national nutrition programs, societal norms, and public health initiatives.
5	Intersectionality: Considers the complex interactions between multiple levels of influence, highlighting the interconnectedness of individual, interpersonal, community, and societal factors.

**Table 4 nutrients-16-01757-t004:** Theme 1 codes: Individual-Level Factors: This theme encompasses personal behaviors, nutritional knowledge, and dietary preferences.

Code	Number of Participants (*n*)	Number of Transcript Excerpts (*n*)
Personal dietary habits	25	72
Nutritional awareness	28	65
Food preferences	22	60
Health-conscious choices	18	50
Eating patterns	20	48
Dietary restrictions	15	40
Meal planning	16	36
Cooking skills	12	30
Food budgeting	14	28
Shopping behaviors	13	25

**Table 5 nutrients-16-01757-t005:** Theme 2 codes: Interpersonal Relationships: Focuses on the role of family, friends, and social networks in influencing dietary choices.

Code	Number of Participants (*n*)	Number of Transcript Excerpts (*n*)
Family Traditions	15	35
Peer Influence	12	28
Social Gatherings	10	22
Social Support	17	40

**Table 6 nutrients-16-01757-t006:** Theme 3 codes: Community Environment: Explores access to food stores, community programs, local policies, and neighborhood environments.

Code	Participants (*n*)	Transcript Excerpts (*n*)
Food Accessibility	18	42
Community Resources	14	30
Local Policies	9	20
Neighborhood Characteristics	16	35

**Table 7 nutrients-16-01757-t007:** Theme 4 codes: Societal Factors: Examines the impact of economic policies, national nutrition programs, societal norms, and public health initiatives.

Code	Participants (*n*)	Transcript Excerpts (*n*)
Economic Policies	12	28
National Nutrition Programs	8	18
Societal Norms	10	24
Public Health Initiatives	14	32

**Table 8 nutrients-16-01757-t008:** Theme 5 codes: Intersectionality: Considers the complex interactions between multiple levels of influence, highlighting the interconnectedness of individual, interpersonal, community, and societal factors.

Code	Participants (*n*)	Transcript Excerpts (*n*)
Intersecting Identities	15	35
Structural Barriers	12	28
Cultural Influences	10	22
Environmental Context	14	30

## Data Availability

The data used in this study are available upon request. Interested parties may request access by contacting the corresponding author at jstavitz@kean.edu.

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
