# Peer review of "Understanding Micronutrient Access through the Lens of the Social Ecological Model: Exploring the Influence of Socioeconomic Factors—A Qualitative Exploration"

_nutrients, 2024, doi:10.3390/nu16111757_

Round 1
Reviewer 1 Report
Comments and Suggestions for Authors
Dear Author
Please find my comments attached.
Kind regards

Reviewer 2 Report
Comments and Suggestions for Authors
The survey carried out is described in detail, the methodology and sampling is presented in great detail with attention to all stages. However, in the reviewer's opinion, it is not possible to draw general conclusions based on the results obtained on such a limited sample of respondents.
The problem is not only the small sample size (only 30 people) but also the large age diversity (aged 20-65). In addition, the author included 1 participant (3.3%) over 70 years old in the results. The inclusion criteria presented indicated an age range of up to 65 years old.
The wide divergence of the respondents does not allow conclusions to be drawn about the characteristics presented. If it is irrelevant from the researcher's point of view then the description in the results section should be changed and socioeconomic data should not be given.
The results obtained may be a pilot study and only as such can they be presented and published, but this still requires the author to change the description of the methodology and correct the title. The reviewer feels that the paper is too long and the chapter on results should be shortened.
